# Discovery of Tryptanthrin and Its Derivatives and Its Activities against NSCLC In Vitro via Both Apoptosis and Autophagy Pathways

**DOI:** 10.3390/ijms24021450

**Published:** 2023-01-11

**Authors:** Yayu Zou, Guanglong Zhang, Chengpeng Li, Haitao Long, Danping Chen, Zhurui Li, Guiping Ouyang, Wenjing Zhang, Yi Zhang, Zhenchao Wang

**Affiliations:** 1College of Pharmacy, Guizhou University, Guiyang 550025, China; 2State Key Laboratory Breeding Base of Green Pesticide and Agricultural Bioengineering, Key Laboratory of Green Pesticide and Agricultural Bioengineering, Ministry of Education, Guiyang 550025, China; 3Guizhou Engineering Laboratory for Synthetic Drugs, Guizhou University, Guiyang 550025, China

**Keywords:** tryptanthrin, tryptanthrin derivatives, autophagy, NSCLC, anticancer

## Abstract

In this study, a series of novel tryptanthrin derivatives were synthesized and their inhibitory activities against selected human cancer cell lines, namely, lung (A549), chronic myeloid leukemia (K562), prostate (PC3), and live (HepG2), were evaluated using a methyl thiazolyl tetrazolium colorimetric (MTT) assay. Among the tested compounds, compound **C1** exhibited a promising inhibitory effect on the A549 cell line with an IC_50_ value of 0.55 ± 0.33 µM. The observation of the cell morphological result showed that treatment with **C1** could significantly inhibit the migration of A549 cells through the cell migration assay. Moreover, after treatment with **C1**, the A549 cells exhibited a typical apoptotic morphology and obvious autophagy. In addition, the detection of apoptosis and the mitochondrial membrane potential indicated that **C1** induced A549 cell apoptosis via modulating the levels of Bcl2 family members and disrupted the mitochondrial membrane potential. Compound **C1** also suppressed the expression of cyclin D1 and increased the expression of p21 in the A549 cells, inducing cell cycle arrest in the G2/M phase in a dose dependent manner. The further mechanism study found that **C1** markedly increased the transformation from LC3-I to LC3-II. Taken together, our results suggest that **C1** is capable of inhibiting the proliferation of non-small cell lung cancer (NSCLC) cells, inducing cell apoptosis, and triggering autophagy.

## 1. Introduction

Tryptanthrin (indolo [2,1-b]quinazolin-6,12-dione) is a natural indole quinazoline alkaloid that is isolated from Chinese medicinal plants such as Strobilanthes cusia, Polygonum tinctorium, and Isatis tinctorial [1]. It is characterized by an indolo [2,1-b]quinazoline core, which exhibits a wide range of biological and pharmacological activities, including anti-tumor [2,3,4], anti-inflammatory [5], anti-tubercular [6], antimicrobial [7,8] and antiviral [9] properties. In artiular, in recent years, the anti-tumor activities of tryptanthrin have attracted great attention from researchers. Furthermore, tryptanthrin has proliferation-attenuating and apoptosis-inducing effects on human CML cells [10], and exerts an anti-breast cancer activity through modulating inflammatory TME both in vitro and in vivo [2]. In addition, researchers found that tryptanthrin can reverse doxorubicin resistance to the breast cancer cell line MCF-7 by inhibiting the expression of the multi-drug resistance (MDR)1 gene [11].

Meanwhile, other indoloquinazolines such as phaitanthrin A, phaitanthrin C, and Candidine [12,13,14] have been reported to have excellent properties, including antitumor, antimicrobial, and anti-inflammatory (Figure 1). Kawakami [15] found that the A-ring compound of 2-(*N*,*N*-dimethyamino) tryptanthrin (Figure 1, Compound **1**) showed remarkable photophysical properties such as emission in the visible region and a wide wavelength absorption. The work of Catanzaro indicated that D-ring 7-substituted derivatives (Figure 1, Compound **2**) exhibited a higher topoisomerase activity than the template compound tryptanthrin [16]. In the process of searching for antioxidants from marine organisms, 4,8,9-trihydroxytryptanthrin (Figure 1, Compound **3**), the first compound found in marine invertebrates, was isolated from the Caribbean ophiuroid [17]. Hence, it can be imagined that the combination of tryptanthrin with other quinazolinones and indoles is able to enhance the pharmacological properties.

In this paper, we changed the 2-position substituents of tryptamine A ring, the 8-position substituents of D ring, and the 4-aza tryptamine derivatives of A ring, respectively, to evaluate the impact on structure−activity relationship (SAR) of different substitutions showing inhibitory activities against four human cancer cell lines, namely lung (A549), chronic myeloid leukemia (K562), prostate (PC3), and live (HepG2) (Figure 2). One of the target compounds **C1** displayed the most promising cytotoxic activity against A549 cancer cells, and we also evaluated the mechanism of cell death. Furthermore, the cellular proliferation, apoptosis, cell cycle, and mitochondrial membrane potential were studied to analyze the underlying mechanism of **C1** against A549 cells. The results suggested that it could induce autophagy in A549 cells. Overall, our study aimed to investigate the role of autophagy in anticancer process in NSCLC.

## 2. Results

### 2.1. Chemistry

The key intermediate compounds **3a–i** and **4a–i** were obtained according to the reported procedures [18,19,20]. The target compounds **A1**–**A9** of A-ring substituted tryptanthrin were synthesized by the optimized Bergman condensation reaction of indigo **3a** and substituted indigo anhydride **4a–i**. The target compounds **B1**–**B9** of D-ring substituted tryptanthrin were synthesized as **A1**–**A9** by the same reaction processes by using substituted indigo **3a–i** and indigo anhydride **4a**. The 4-azatryptanthrin derivatives were synthesized by anthranilic acid, substituted indigo in the presence of DBU, *N*-methylmorpholine, and HBTU with DMF as a solvent at room temperature. In order to obtain the aliphatic amine substituted azatryptanthrin compounds, we further derived 4-azachlorotryptamine **C3** with aliphatic amine compounds to obtain the title compounds **D1**–**D5** (Figure 3). All structures of the title compounds **A1**–**A9**, **B1**–**B9**, **C1**–**C5**, and **D1**–**D5** were confirmed by ^1^H NMR spectroscopy, ^13^C NMR spectroscopy, and HR-MS spectroscopy (Appendix A).

### 2.2. Anti-Proliferative Activity

The anti-proliferative activity of tryptanthrin derivatives in vitro was evaluated in four tumor cell lines (A549, K562, PC3, and HepG2) using an MTT assay. The inhibitory activity of all of the target compounds, the reference standard tryptanthrin (TRYP), and 5-fluorouracil (5-Fu) are listed in Table 1, all of which were at a fixed concentration of 10 μM. Among the four series compounds, the **C** series compounds (**C1**–**C5**) exhibited the best anti-tumor activity against A549 cells, and their activities were better than that of the positive control drugs 5-fluorouracil and tryptanthrin. Furthermore, **B** series compounds (**B-1**, **2**, **3**, **7**, **8**, and **9**) and **D** series compounds (**D1**–**D5**) also showed good antiproliferative activities. The above results indicate that the antitumor activity of all of the target compounds on A549 cells was superior to that of the other three tumor cells.

To further explore the SARS of the synthesized compounds, some potential compounds were chosen to evaluate the concentrations that caused a 50% inhibition rate of cancer cell growth (IC_50_) on the studied tumor cell lines (Table 2). As summarized in Table 2, the leading compounds **C1**–**C5** displayed an excellent inhibition activity against A549 cancer cell lines, with IC_50_ values ranging from 0.55–1.29 µM. However, **D3** and **D4** showed lower IC_50_ values than **C1**–**C5** toward A549 cells, which might be as a result of the introduction of the fatty amine group in the 10th position of **C1** that increased its steric hindrance. Furthermore, the introduction of halogen group (F, Cl) into the **D** ring of tryptamine was able to greatly improve the anticancer activity, and the double F substituent or the nitro group could further enhance its activity due to the strength of the electron-withdrawing group. In addition, the activity of the **A** series was less than satisfactory, which indicates that the activity of replacing it on the **D** ring in tryptamine was generally better than that on the **A** ring (Figure 4). According to the superior activity and selectivity results, compound **C1** was selected as the candidate for further mechanism research. Therefore, the inhibitory influence of **C1** on the proliferation of the normal cell line HEK-293 was determined using the MTT assay. The data indicate that **C1** showed a low toxicity to human normal cells (Appendix A).

### 2.3. Cell Morphological Analysis

To monitor the effect of **C1** on cell proliferation, we investigated the impact of **C1** on A549 cells using a wound healing assay. Cell migration and motility are usually related to the metastatic activity of cancer cells. As shown in Figure 5A, compared with the control group, a significant decrease in migration in the medication administration groups was observed. From the cell quantification result in Figure 5B, we found that **C1** could effectively inhibit the migration of A549 cells in a concentration-dependent manner. A more prominent effect was detected when the concentration was 1 μM, and only 15% of cells migrated after 24 h.

Furthermore, to further investigate the mechanism of A549 cell growth inhibition, Hoechst 33,258 staining and TEM were carried out to observe the morphological changes of A549 cells that were exposed to **C1** in different concentrations for 48 h. It can be seen from Figure 5D that the control group of A549 cells exhibited small and round normal nuclei. In contrast, when treated with 0.25 and 0.5 µM **C1**, the early signs of apoptosis were characterized by condensed chromatin and cell membrane blebbing. Meanwhile, with the increase in concentration, chromatin shrinking and gathering towards the nuclear membrane were observed in the cells. The nucleus was disintegrated to form fragments, and the fluorescence generated by the binding of Hoechst 33,258 to the DNA was also enhanced. Furthermore, it was also found that the apoptosis and autophagosomes were observed in the 2 µM **C1** group compared with the control group (Figure 5C). Taken together, all of the experimental phenomena indicate that the anti-proliferative activity of **C1** on cells stemmed from its abilities to resist migration and promote programmed cell apoptosis and autophagy.

### 2.4. Cell Apoptosis Analysis and Effect on the Mitochondrial Membrane Potential (DΨm)

To quantify the effects of **C1** on cell apoptosis, A549 cells were treated with different concentrations for each preparation, and then apoptosis was investigated by flow cytometry using the annexin V-FITC/PI apoptosis assay detection kit. Figure 6 shows the analysis results of A549 cells cultured with **C1** for 48 h. As shown in Figure 6A, few necrotic or apoptotic cells were detected in the control group. In comparison with the control group, the total apoptosis rates (early and late apoptotic cells) of the A549 cells treated with **C1** in 0.25, 0.5, 1, and 2 μM, were 75.02%, 78.82%, 92.89%, and 97.53%, respectively, which suggests that **C1** induced A549 cell apoptosis in a dose-dependent manner, and the effect of late apoptosis was obvious. Meanwhile, it was observed that **C1** could significantly reduce the expression of the Bcl-2 proteins and increase the expression of Bax, which resulted in an increase in the pro-apoptotic Bax/antiapoptotic Bcl-2 ratio (Figure 6C).

As the disruption of the mitochondrial membrane integrity, as well as loss or collapse of the mitochondrial membrane potential, became hallmarks in the early cell apoptosis events, we tested this possibility by examining the influence of **C1** on DΨm of the A549 cancer cell line using rhodamine-123 staining, a green-emitting cationic fluorescent dye. The results displayed in Figure 6E clearly show that the mitochondrial membrane potential dropped sharply and even collapsed at a concentration of 2 μM for **C1**, whereas these remarkable features were almost absent in the untreated group or the other concentration treated groups (Figure 6(Ea–e)). An additional experiment was conducted to exclude that the drastic effect at 2 μM was caused by the side effects of **C1**. As shown in Appendix A, the normal cell line of HEK-293 was treated with **C1**, and the cell proliferation and death were observed by DAPI staining. The results suggest that the toxic effects and side effects of **C1** were perfectly general and acceptable. The drastic effect in this study was likely to be caused by excessive autophagy of the cells, because autophagy also occurred at 2 μM which was observed by an electron microscope, rather than the side effects of **C1** (Figure 5C). Based on the above results, cell apoptosis was a normal process when the concentrations were in the range of 0.25–1 μM; however, the collapse of the mitochondrial membrane potential and excessive autophagy led to apoptosis when treated with 2 μM of **C1**.

### 2.5. Cell Cycle Analysis

To further explore the anticancer properties of **C1**, we evaluated the influence of **C1** on the cell cycle distribution. The cell cycle phase distribution was analyzed by flow cytometry. As shown in Figure 7A,B, treatment of A549 cells with **C1** (0.25, 0.5, 1, and 2 μM) for 24 h increased the percentage of cells in G2/M (Figure 7A,B). In addition, the impacts of **C1** on the expression of the cell cycle-related proteins were analyzed by Western blot. **C1** significantly down regulated the expression of Cyclin D1 and upregulated the expression of the CDK inhibitor p21 (Figure 7C,D). These results reveal that **C1** also exerted anti-tumor properties by affecting the cycle distribution.

### 2.6. Effect of C1 on Expressions of Autophagy Related Protein LC3

In the previous results, it was mentioned that autophagy could be observed using a transmission electron microscope. Meanwhile, LC3-II was another important marker of autophagosomes, which was increased in the autophagosome membrane [21]. The expression changes of LC3 in the A549 cells could be detected by Western blot after treatment with **C1** for 48 h, and the transformation from LC3-I to LC3-II increased significantly (Figure 8A,B). The results showed that **C1** was able to change the expression of autophagy-related proteins in A549 cells.

## 3. Discussion

Autophagy is a key molecular pathway for maintaining cellular and organismal homeostasis [22], promoting cell survival by removing the damaged organelles and macromolecules from the cells and maintaining cellular homeostasis [23,24]. However, autophagy can also promote cell death, which is associated with apoptosis [25]. There is a strong link between autophagy and apoptosis: autophagy and apoptosis often interact with each other, where autophagy can promote or inhibit apoptosis, and similarly apoptosis is able to facilitate or hinder autophagy as well [26,27]. In recent years, a large number of studies have noted that abnormal cellular autophagy is usually related to the occurrence of various tumors [23,28]. Moreover, the disordered autophagy process leads to various diseases, such as injury, infection, and cancer [29]. Lung cancer remains one of the most common malignant tumors and has become the leading cause of cancer death worldwide [30]. Among all of the lung cancers, non-small cell lung cancer (NSCLC) is the main type, accounting for nearly 85% of the total lung cancers [31,32]. Autophagy dysfunction was found to be involved in NSCLC regulation, which played a positive or negative role in promoting the apoptosis of NSCLC cells. Generally, inhibiting autophagy can limit the ability of cells to overcome stress and maintain homeostasis [33]. However, in some cases, autophagy enhances cell death. Shipeng et al. [34] indicated that an inflammatory microenvironment-induced autophagy promoted the apoptosis of mesenchymal stem cells, and Mutian et al. [35] found that a botanical hormone induced apoptosis and pro-apoptotic autophagy in NSCLC cells through the ROS pathway. Thus, autophagy is a fundamental determinant of human health, and autophagy-regulating interventions emerge as promising approaches to prevent or mitigate phenotypic abnormalities in most common human diseases.

This study was conducted to assess the antitumor activity of tryptanthrin derivatives in NSCLC A549 cancer models in vitro, and to explore the important role of autophagy caused by tryptanthrin in the tumor cells. Autophagy is a major intracellular degradation system, and its degradation ability originates from lysosomes [36], which is considered as a process of regulating cancer. The new role of autophagy in regulating early endosomal homeostasis is well known. Furthermore, autophagy also plays a key role in the pathogenesis, survival, and response to the treatment of all cancers by inhibiting the occurrence or maintaining the growth and survival of established tumors [37,38]. Generally, basic autophagy can damage cancer progression under stress; nevertheless, an abnormal autophagy activity was found to facilitate the survival of cancer cells [39]. Remarkably, a large number of studies have shown that tumors are more dependent on autophagy than normal tissues, which indicates that inhibiting or activating autophagy might be a feasible way to treat cancers [40]. Currently, apoptosis is well known as the major and most-described form of programmed cell death, and it plays an indispensable role in cancer therapy [41]. Therefore, both apoptosis and autophagy are involved in determining the fate of cancer cells, and there are interactions between them. Fang et al. [42] showed that inhibiting Ulk1 impeded the growth of NSCLC cells by regulating both autophagy and apoptosis pathways. The epidermal growth factor receptor (EGFR) is a well-studied receptor tyrosine kinase that plays a vital role in regulating organ development and cancer progression [43], especially in NSCLC. Thus, the effects of autophagy on the EGFR signaling pathway were also briefly discussed in this study.

In the current study, we first demonstrated the anti-proliferation effects and clarified the possible molecular mechanisms of **C1** on NSCLC. To evaluate the mechanism of cytotoxicity, apoptosis, and cell cycle during the treatment, further experiments on A549 cells were carried out, including MTT assays and flow cytometry. In particular, the results showed that **C1** induced apoptosis in A549 cells in a dose-dependent manner via up-regulating the levels of Bax and reducing the expression of the anti-apoptotic protein Bcl-2. The mitochondrial membrane potential revealed that **C1** substantially decreased the levels of the mitochondrial membrane potential at a concentration of 2 μM. Meanwhile, it could arrest the A549 cell cycle in the G2/M phase by down-regulating the expression of Cyclin D1, and this effect was also associated with the up-regulation of p21. Furthermore, the cell migration experiment showed that **C1** affected the cell motility in A549 cells. Autophagy, as it is known, is important to maintain early endosome functions. Interestingly, it was found that **C1** could cause the autophagy of cells through electron microscope observation. The further exploration of the impact of **C1** on the expression of the LC3 protein that was related to autophagy revealed that the critical autophagy formation protein LC3-II was significantly over expressed when treated with **C1** at a concentration of 1 μM. More interestingly, the protein expression of the tumor suppressor genes p-EGFR and p-Akt was markedly up-regulated, and the protein expression of EGFR and Akt did not change much in A549 cells treated with **C1** (Appendix A). Based on the above results, our findings suggest that **C1** inhibits the growth of NSCLC cells via apoptosis and autophagy, and the occurrence of autophagy may also affect the signal transduction of EGFR.

More importantly, a successful candidate drug should have a good ADME profile, which includes absorption, distribution, metabolism, and excretion [44,45,46]. In this study, the ADME profiles of compound **C1** were calculated using Swiss ADME software and ADMETlab 2.0 online websites. Unfortunately, the target compound **C1** only displayed appropriate ADME properties (Appendix A). Our research group will further optimize the structure of lead compounds and explore their mechanisms in the future.

## 4. Materials and Methods

### 4.1. Chemistry and Instruments

All of the reagents were purchased from commercial sources. All of the solvents and reagents were reagent-grade without further purification. NMR data were obtained by Bruker Ascend 400 and 500 MHz (Bruker Optics, Fällanden, Switzerland). High resolution mass spectrometry (HRMS) spectra were conducted using a Thermo Scientific Q Exactive (Thermo Scientific, St. Louis, MO, USA). The melting points (m. p.) for all of the title tryptanthrin derivatives were detected using a (X-4D) digital micro melting point apparatus.

### 4.2. Pharmacology

#### 4.2.1. Cell Culture and Treatment

Lung (A549), chronic myeloid leukemia (K562), prostate (PC3), and live (HepG2) cells were purchased from the Cell Bank of the Chinese Academy of Sciences (Kunming, China). The A549, K562, and PC3 cells were cultured in an RPMI-1640 medium and the live HepG2 cells were cultured in a MEM medium. All of the mediums were supplemented with 10% FBS, 100 μg/mL penicillin, and 100 μg/mL streptomycin. All of the cells were cultivated in 5% CO_2_ incubator at 37 °C.

#### 4.2.2. MTT Assay

To investigate the anticancer activity of 4-aza-tryptanthrin derivatives, A549, K562, PC3, and HepG2 cells were treated with target compounds, respectively, for 48 h in a range of concentrations (from 0.625 to 10 μM) and were subjected to a proliferation assay using an MTT assay. Exponentially growing cells were seeded into each well of 96-well plates containing 20 μL of fresh growth medium, allowed to attach for 24 h, and were then subjected to target compounds (0.625–10 μM). After incubation for 48 h, the viable cells were stained with MTT (0.5 mg/mL) for 4 h, and the produced formazan crystals were dissolved with 150 µL of dimethyl sulfoxide. Absorbance was measured at 490 nm using a Titertek multiskan automated microplate reader (Tecan, Infinite M200 Pro). The relative cell viability was calculated according to the following equation:$$\mathrm{Cell}\;\mathrm{viability}\left(\%\right)=\frac{\mathrm{experimental}\;\mathrm{OD}\;\mathrm{value}}{\mathrm{control}\;\mathrm{OD}\;\mathrm{value}}\times 100$$
$$\mathrm{Cell}\;\mathrm{inhibition}\left(\%\right)=\frac{\mathrm{control}\;\mathrm{OD}\;\mathrm{value}-\mathrm{experimental}\;\mathrm{OD}\;\mathrm{value}}{\mathrm{control}\;\mathrm{OD}\;\mathrm{value}}\times 100$$

#### 4.2.3. Cell Migration Assay

We evaluated the inhibition of the migration of **C1** on A549 cells using a wound healing assay. Wounds were created on a confluent cell monolayer culture of A549 cells by using a sterile 10 μL pipette tip and were treated with 0.25, 0.5, and 1 μM of **C1**. The migration of A549 cells was recorded by microscopic observations at 0 h, 12 h, and 24 h.

#### 4.2.4. Hoechst33258 Staining

The A549 cells in the logarithmic growth phase were cultured in six-well plates (1 × 10^6^ cells/well) and treated with 0.25, 0.5, 1, and 2 μM of **C1** or without **C1** (control (DMSO)). After 48 h of various concentrations of **C1** treatment, the cells were washed two times with PBS, followed by fixation with 4% paraformaldehyde and added with 1 mL/well of dye for 15 min. The cells were observed for apoptotic characteristics under a fluorescence microscope.

#### 4.2.5. Transmission Electron Microscopy

The cells were collected and 3% glutaraldehyde fixation was used to fix the sample overnight, then fixation was followed by 3–5 min washes with 0.1% sodium cacodylate buffer, at pH 7.4. The cells were post-fixed with a solution containing 1% osmium tetroxide and 2% K_4_Fe, stained with 1% uranyl acetate, and pelleted in 2% agar. The pellets were dehydrated in graded ethanol solution and embedded in Ep812. Ultrathin (60–90 nm) sections were cut on a microtome, collected on Rhodanimu 400-mesh grids, post-stained with uranyi acetate and lead citrate, and washed with water. The sections were examined in a transmission election microscope (JEM-1400FLASH).

#### 4.2.6. Annexin V-FITC/Propidium Iodide Dual Staining Assay

The activation of apoptosis plays an important role in treatment-induced cancer cell death. To investigate whether apoptosis could underlie the anticancer activity of **C1**, A549 cells were treated with **C1** (0.25, 0.5, 1, or 2 μM) for 48 h and the percentage of apoptotic cells was determined by flow cytometry. The cells were collected using trypsinization. The cells (1 × 10^6^/mL per well) were plated in six-well culture plates and allowed to grow for 24 h. After treatment with 0.25, 0.5, 1, and 2 μM of **C1** for 48 h, the cells were collected by trypsinization (0.05% strypsin, 0.02% EDTA). The collected cells were washed two times with ice-cold PBS, counted, and 100 μL of cell suspension was prepared with a density of 1 × 10^6^ cells/mL, and then incubated with 5 μL Annexin V-FITC and 5 μL propidium iodide (PI) in the dark for 15 min. After 15 min of incubation and 400 μL of 1× binding buffer being added, the cells were detected using a flow cytometer (FAC Scan, Becton Dickenson, Franklin Lakes, NJ, USA). The system software (Cell Quest BD Biosciences, San Jose, CA, USA) was used to analyze the apoptosis experiment.

#### 4.2.7. Measurement of Mitochondrial Membrane Potential (DΨm)

The A549 cells (5 × 10^5^ cells/well) were plated in a six-well culture plate. After leaving it overnight, the cells were treated with 0.25, 0.5, 1, and 2 μM of **C1** for 24 h. After treatment, the adherent cells were collected by trypsinization, washed with PBS, and resuspended in a solution of PBS containing rhodamine-123 (5 mg/mL) and additionally incubated at room temperature for 30 min. The cells were washed two times with PBS to remove excess dye and resuspended in PBS. Then, 3000 cells from each sample were analyzed using a flow cytometer (FAC Scan, Becton Dickenson) forrhodamine-123 fluorescence.

#### 4.2.8. Cell Cycle Analysis

To analyze the cell cycle, A549 cells were stained using a PI apoptosis detection kit. The A549 cells were seeded on six-well plates and allowed to attach for about 24 h. Then, **C1** (0.25, 0.5, 1 and 2 μM) or the control were added to the cells for another 24 h. Then, the cells were collected using trypsinization (0.05% trypsin, 0.02% EDTA), washed three times with PBS, and fixed with 70% ethanol at 4 °C overnight. After being fixed, the cells were again washed with PBS and incubated at 37 °C with 100 μL RNase A for 30 min. After this, cells were incubated with 400 μL PI for 30 min at 4 °C. After staining, cells went through the flow cytometry (FAC Scan, Becton Dickenson) using Cell Quest software and recording PI in the FL2 channel.

#### 4.2.9. Western Blotting Analysis

After drug treatment, the cells were lysed in a 1× RIPA lysis buffer with proteinase and phosphatase inhibitors for 15 min on ice, scraped off from the plate, and the cell lysates prepared. The lysates were centrifuged at 1.2 × 10^4^ rpm at 4 °C for 15 min to collect the supernatants and the protein concentration of each sample was quantified using the BCA protein assay kit. Protein was separated by SDS-PAGE and then transferred onto the PVDF membrane. After blocking with 5% skim milk for 2 h, they were subsequently incubated with primary antibodies at 4 °C overnight and then with HRP-labeled secondary antibodies for 2 h at room temperature. Finally, the bands were visualized using an enhanced chemiluminescence system (Millipore, Burlington, MA, USA).

#### 4.2.10. Statistical Analysis

Statistical processing of the results was performed using GraphPad Prism 8.0 software (GraphPad Inc., San Diego, CA, USA). All data were expressed as mean ± SD and all the data provided have been verified by at least three independent experiments. Differences among groups were considered significant at *p* ≤ 0.05.

## 5. Conclusions

In conclusion, the comprehensive evaluation of the pharmacological effects of a number of novel tryptanthrin derivatives on tumor cells were performed in vitro. The antiproliferative activity of the products was screened against A549, K562, PC3, and HepG2 cell lines. Most derivatives showed a moderate to significant potency on the tested tumor cells, and the most promising **C1** (IC_50_ = 0.55 ± 0.33 µM) exhibited an excellent anti-proliferative activity against A549. Firstly, the SAR study indicated that replacing the A ring of tryptamine with the pyridine ring might make great contributions to the antitumor activity. Secondly, the anti-tumor activity of introducing an electron-withdrawing substituent into the B ring was better than that of introducing a fatty amine group. Furthermore, the anti-tumor mechanism of the representative compound **C1** was represented in Figure 9. **C1** could downregulate cell proliferation, and induce apoptosis and cell cycle arrest in A549 cells. Most importantly, we found that **C1**-induced apoptosis was caused by autophagy, which in turn influenced the transmission of the EGFR signal pathway. From these results, **C1** is expected to be a promising anti-lung cancer chemotherapeutic drug for further evaluations.

## Figures and Tables

**Figure 1 ijms-24-01450-f001:**
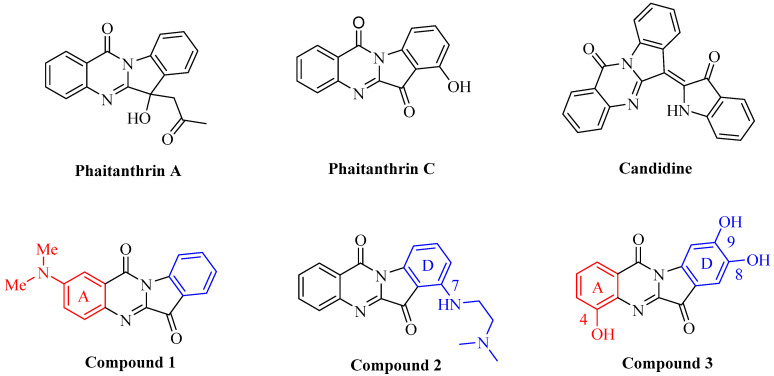
Some biologically active indoloquinazolines of natural alkaloids and structures of small molecule compounds with wide applications.

**Figure 2 ijms-24-01450-f002:**
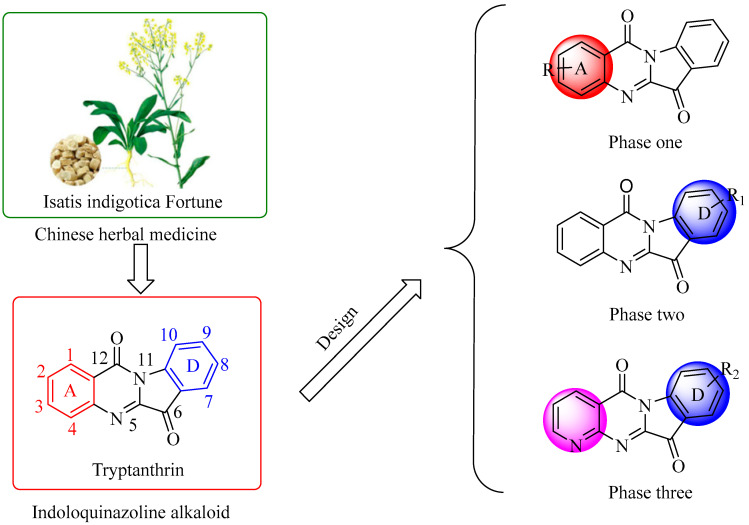
Strategy of tryptanthrin and its derivatives as potent antitumor activity agents.

**Figure 3 ijms-24-01450-f003:**
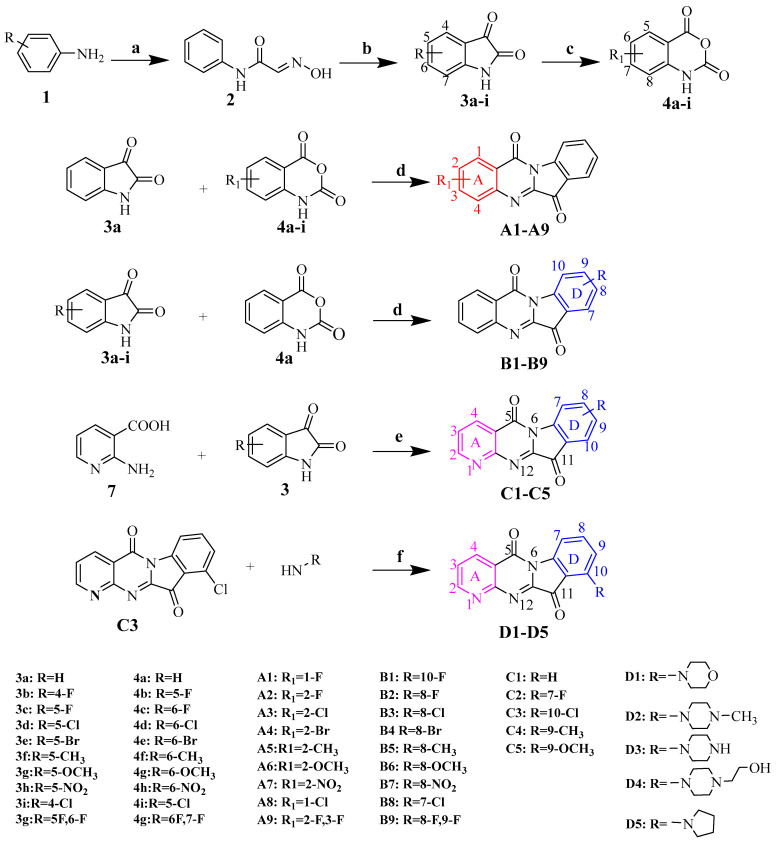
Synthetic route of tryptanthrin derivatives containing A ring, D ring, and azatryptanthrin, respectively. Reagents and conditions: (**a**) NH_2_OH, CHCl_3_, Na_2_SO_4_, reflux; (**b**) H_2_SO_4_, 65 °C; (**c**) m-CPBA, CH_2_Cl_2_, room temperature; (**d**) toluene, TEA, reflux; (**e**) 1. DBU, DMF, room temperature; 2. NMM, HBTU, room temperature; and (**f**) NMP, K_2_CO_3_.

**Figure 4 ijms-24-01450-f004:**
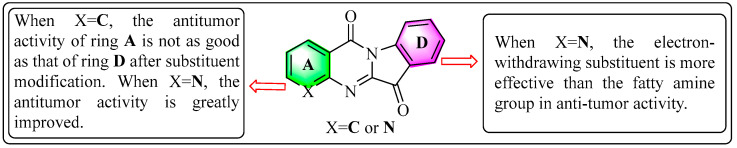
The structure–activity relationship of the tryptanthrin derivatives.

**Figure 5 ijms-24-01450-f005:**
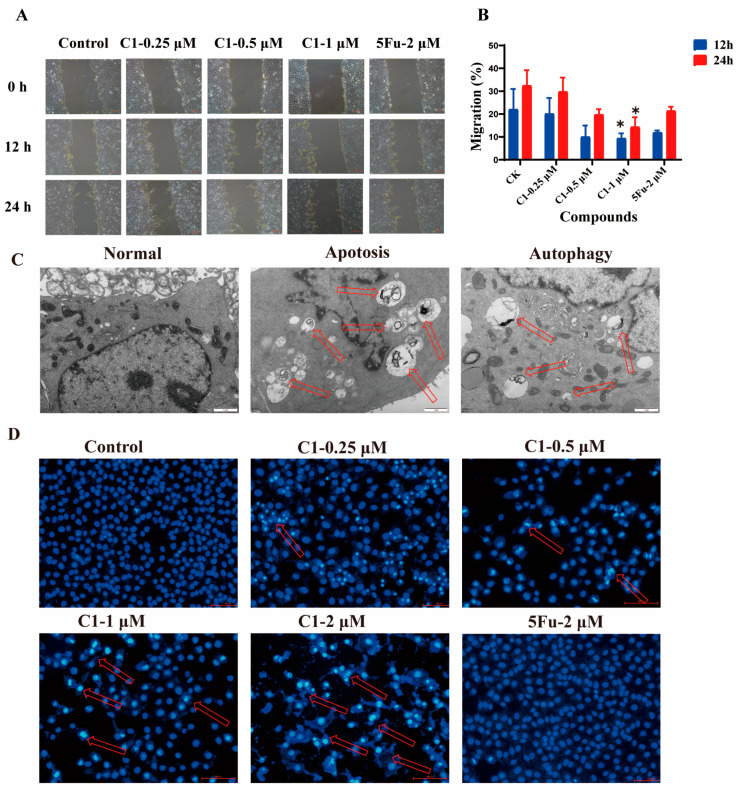
Compound **C1** inhibited NSCLC cell proliferation and migration. (**A**) Cell migration assay. The images are captured using phase contrast microscopy at 0 h, 12 h, and 24 h. Scale bars are 100 μm. (**B**) The relative percentage of cells that migrated are quantified using Prism Elements software. All data presented are mean ± SD of three similar independent experiments. * Statistically significant (*p* < 0.05). (**C**) Electron microscopy analysis of apoptosis and autophagy after the cells were treated with **C1** (2 μM) for 48 h. The arrow indicates apoptotic bodies and the autophagosome or autolysosome. Scale bars are 1 μm. (**D**) Hoechest 33,258 staining shows an apoptosis effect. The cell morphology is observed under the fluorescence microscope (20×). Scale bars are 100 μm.

**Figure 6 ijms-24-01450-f006:**
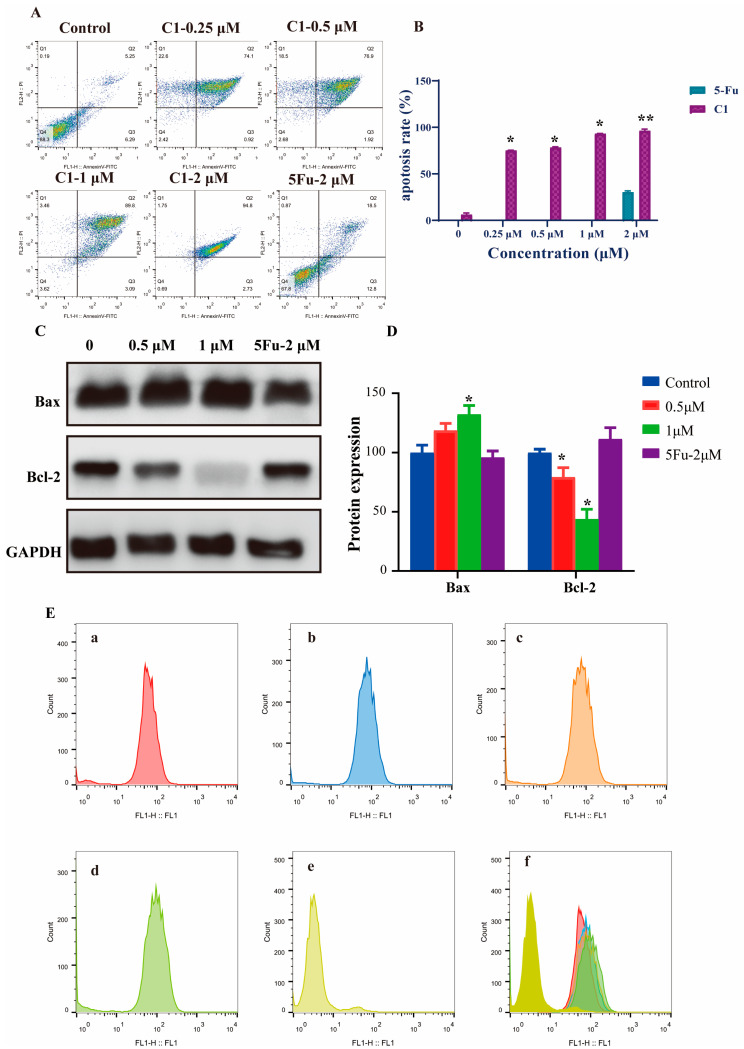
Effects on the apoptosis and mitochondrial membrane potential analyzed by flow cytometry. (**A**) Flow cytometry profiles indicate that **C1** dose-dependently promoted A549 cell apoptosis. (**B**) Statistics show that **C1** caused an apoptosis rate of A549 cells. (**C**) After the cells are treated with **C1** (0.5 and 1 μM) for 48 h, expressions of the related apoptotic proteins Bax and Bcl-2 are performed by Western blot, and GAPDH is used as a loading control. (**D**) Relative levels of the Bax and Bcl-2 protein are detected using Western blot analysis. Data are expressed as the mean values ± standard deviations (SD) of three different experiments. * Statistically significant (*p* < 0.05). ** Statistically extremely significant (*p* < 0.01). (**E**) **C1** triggers a change in mitochondrial membrane potential (DΨm). A549 cells treated with (**a**) control (DMSO), (**b**) 0.25 μM, (**c**) 0.5 μM, (**d**) 1 μM, and (**e**) 2 μM **C1,** and (**f**) an overlay of (**a**–**d**) processed by rhodamine-123 staining followed by flow cytometry analysis.

**Figure 7 ijms-24-01450-f007:**
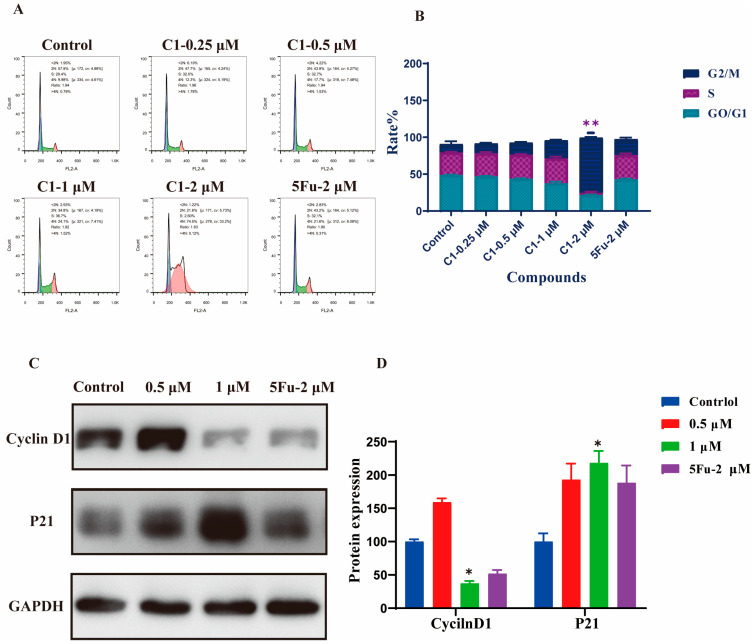
The effects of the cell cycle affected by **C1** in A549 cells. (**A**) The flow cytometry analysis indicates that **C1** dose-dependently promotes the A549 cell cycle. (**B**) Statistics show that **C1** causes the cell cycle of A549 cells. (**C**) Effects of **C1** on cell-cycle-related proteins Cyclin D1 and P21 are analyzed after exposing the cells to **C1** at concentrations of 0.5 and 1 μM for 48 h. GAPDH is used as a loading control. (**D**) Relative levels of the Cyclin D1 and P21 protein are detected using Western blot analysis. Data are expressed as the mean values ± standard deviations (SD) of three different experiments. * Statistically significant (*p* < 0.05). ** Statistically extremely significant (*p* < 0.01).

**Figure 8 ijms-24-01450-f008:**
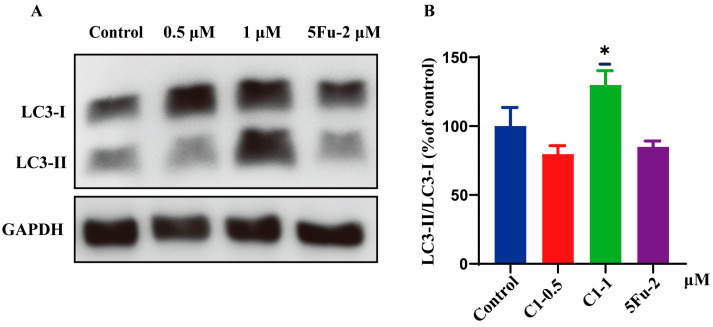
The effects of **C1** on the autophagy-related protein LC3. (**A**) Western blot analysis of LC3-I and LC3-II of A549 cells treated with **C1** (0, 0.5, 1 μM) for 48 h. (**B**) The relative protein expression levels of LC3-I and LC3-II are quantified by normalization to GAPDH. GAPDH is used as a loading control. The values are expressed as mean SD from three individual experiments. * Statistically significant (*p* < 0.05).

**Figure 9 ijms-24-01450-f009:**
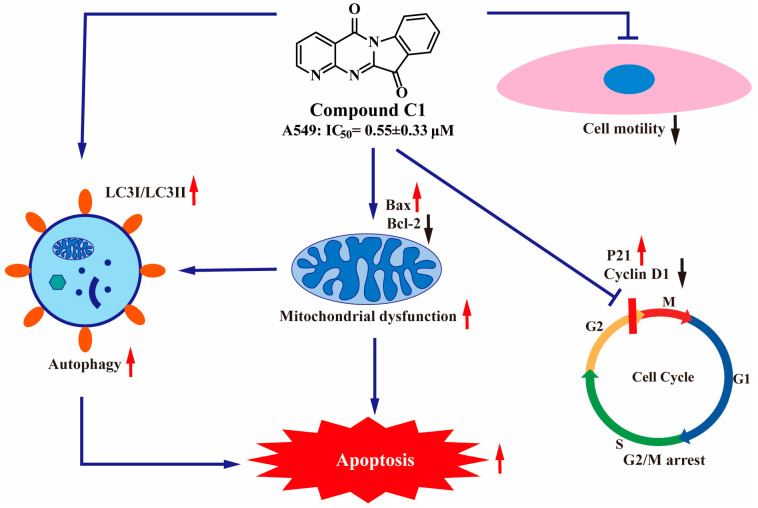
The mechanism of compound **C1**.

**Table 1 ijms-24-01450-t001:** The anti-proliferative activity of tryptanthrin derivatives against cell cancer lines.

Compounds	Inhibition Rate ^1^ (%) ± SD (Concentration: 10 μM)			
	A549	K562	HepG2	PC3
**A1**	73.73 ± 1.45	48.38 ± 0.11	21.22 ± 2.00	61.88 ± 1.97
**A2**	59.26 ± 1.12	46.57 ± 1.37	25.19 ± 3.04	52.87 ± 0.82
**A3**	27.23 ± 0.66	16.64 ± 3.22	19.67 ± 2.38	45.90 ± 2.24
**A4**	36.64 ± 3.53	23.78 ± 0.97	22.21 ± 0.62	34.44 ± 1.61
**A5**	23.55 ± 1.00	25.89 ± 1.53	27.35 ± 2.55	60.00 ± 2.98
**A6**	31.79 ± 2.47	31.25 ± 0.18	17.43 ± 1.22	47.79 ± 1.08
**A7**	27.42 ± 3.02	31.39 ± 1.38	20.09 ± 0.96	69.14 ± 0.50
**A8**	57.43 ± 1.32	24.45 ± 2.34	25.76 ± 1.65	50.89 ± 1.24
**A9**	18.03 ± 1.66	7.84 ± 2.22	20.45 ± 2.88	52.83 ± 3.93
**B1**	67.27 ± 2.79	73.94 ± 0.93	76.39 ± 2.43	66.17 ± 4.20
**B2**	96.22 ± 0.49	88.82 ± 1.23	38.27 ± 1.39	72.52 ± 0.80
**B3**	77.37 ± 0.54	82.44 ± 1.69	23.48 ± 2.82	58.55 ± 2.63
**B4**	57.27 ± 2.54	77.36 ± 0.77	36.26 ± 1.66	64.09 ± 0.43
**B5**	31.79 ± 2.47	26.68 ± 3.09	17.43 ± 1.22	47.79 ± 1.08
**B6**	58.58 ± 1.19	66.96 ± 1.16	29.87 ± 1.75	56.72 ± 1.03
**B7**	91.38 ± 2.74	95.85 ± 0.08	52.52 ± 2.86	81.06 ± 3.96
**B8**	82.58 ± 0.82	44.21 ± 3.94	12.28 ± 0.67	56.73 ± 4.68
**B9**	95.60 ± 0.59	96.25 ± 0.41	15.50 ± 2.36	98.21 ± 0.42
**C1**	96.78 ± 0.59	98.31 ± 0.74	77.19 ± 1.09	92.85 ± 0.79
**C2**	96.33 ± 0.67	98.79 ± 0.14	76.33 ± 2.23	99.32 ± 0.25
**C3**	96.45 ± 1.13	99.20 ± 0.22	82.12 ± 1.50	99.39 ± 0.01
**C4**	96.67 ± 0.57	78.96 ± 1.71	32.95 ± 2.05	81.63 ± 0.03
**C5**	96.64 ± 0.66	98.93 ± 0.29	78.48 ± 3.26	97.52 ± 0.13
**D1**	23.09 ± 4.21	14.25 ± 1.97	17.04 ± 1.12	32.95 ± 8.44
**D2**	16.54 ± 3.83	5.64 ± 1.18	16.54 ± 3.83	32.01 ± 2.66
**D3**	56.38 ± 6.71	42.95 ± 1.19	24.93 ± 0.78	64.50 ± 1.48
**D4**	50.47 ± 3.27	17.65 ± 5.49	21.44 ± 6.74	44.25 ± 3.23
**D5**	22.36 ± 0.62	6.10 ± 1.12	19.29 ± 6.51	25.76 ± 7.78
TRYP ^2^	52.26 ± 0.72	50.85 ± 2.41	42.87 ± 1.60	59.01 ± 2.70
5-Fu ^3^	64.31 ± 2.42	56.38 ± 2.07	33.39 ± 2.42	61.70 ± 3.40

^1^ The anti-proliferation activity of all of the target compounds on cancer cell lines was determined using an MTT assay. The data represent the average of three determinations. ^2^ Tryptanthrin (TRYP) as reference standard. ^3^ 5-Fluorouracil (5-Fu) as the reference standard.

**Table 2 ijms-24-01450-t002:** IC_50_ values (µM) of the tested compounds against the human non-small cell lung cancer A549 cell line.

Compounds	IC_50_ ± SD ^1^ (μM)			
	A549	K562	HepG2	PC3
**A1**	5.45 ± 0.46	6.86 ± 0.10	6.26 ± 0.59	12.36 ± 0.85
**B1**	6.45 ± 1.26	3.69 ± 0.26	5.89 ± 0.48	7.05 ± 1.23
**B2**	2.36 ± 0.80	2.66 ± 0.10	1.97 ± 0.77	10.59 ± 0.55
**B3**	4.21 ± 0.36	3.67 ± 0.39	2.95 ± 0.92	8.48 ± 3.98
**B7**	6.88 ± 3.51	8.05 ± 0.01	7.89 ± 0.58	11.83 ± 0.19
**B8**	2.34 ± 0.65	6.77 ± 0.67	5.63 ± 1.56	9.70 ± 1.66
**B9**	2.15 ± 0.43	3.31 ± 0.23	9.84 ± 0.68	10.44 ± 0.87
**C1**	0.55 ± 0.33	1.02 ± 0.09	0.91 ± 0.79	1.67 ± 0.12
**C2**	1.04 ± 0.04	1.53 ± 0.05	1.85 ± 0.87	1.61 ± 0.05
**C3**	0.90 ± 0.12	0.83 ± 0.10	0.97 ± 0.85	1.07 ± 0.06
**C4**	1.08 ± 0.23	1.97 ± 0.28	1.77 ± 0.95	2.48 ± 0.46
**C5**	1.29 ± 0.31	1.86 ± 0.02	1.33 ± 0.56	1.30 ± 0.01
**D3**	8.12 ± 1.26	5.50 ± 0.47	5.67 ± 0.84	10.13 ± 0.18
**D4**	9.16 ± 0.95	12.85 ± 1.54	10.92 ± 1.84	13.85 ± 0.95
TRYP	8.25 ± 0.69	12.04 ± 0.69	9.51 ± 0.23	14.52 ± 3.29
5-Fu	6.30 ± 1.62	8.09 ± 2.05	8.53 ± 0.95	11.09 ± 1.40

^1^ IC_50_ values are the concentrations that cause 50% inhibition of the cancer cell growth. Data represent the mean values ± standard deviation of three independent experiments performed in triplicate.

## Data Availability

Not applicable.

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
