# Peer review of "Discovery of Tryptanthrin and Its Derivatives and Its Activities against NSCLC In Vitro via Both Apoptosis and Autophagy Pathways"

_ijms, 2023, doi:10.3390/ijms24021450_

Round 1

Reviewer 1 Report

Zou and Zhang et al. developed derivative compounds that show anti-cancer activity based on tryptanthrin. Among them, the compound C1 showed the strongest activity, leading to apoptosis of cancer cell lines. The authors also demonstrated that the apoptosis would be caused by autophagy. Their results will be informative, but before consideration of publication in the International Journal of Molecular Science, the authors should respond to comments described below.

1) The title and contents are misleading. As shown in Tables 1 and 2, the compound C1 is not specific to A549 lung cancer cell line. This compound will affect to cell growth of a lot of types of cancer cell lines. Thus, the manuscript must be re-written without emphasizing lung cancer and EGFR signaling pathway.

2) Related to 1), the results on EGF signaling pathway should be just one aspect among effects of C1. Therefore, it should be discussed in the Discussion section. In this context, Figure 6C and D should be shown in Supplementary Information. If the authors insist lung cancer and EGFR signaling pathway, they should add more convincing results on specific effects of C1 to lung cancer and EGFR signaling pathway.

3) C1 showed no effects at 1 μM but remarkable ones at 2 μM in some cases (e.g., Figures 4A, 4E, and 5A). In this regard, the cellular population at 2 μM are different from those at 0.25 - 1 μM (Figure 4A). Such drastic effects at 2 μM would reflect side-effects of C1. Additional experiments or convincing explanation would be necessary to avoid potential readers’ confusion.

4How about effects of C1 to normal cells?

5) The English writing style is premature. I strongly recommend having a commercially available English editing service.

6) Scheme 1 should be included in a series of Figures.

7) Figure 3C: Concentration of C1 is unclear.

8) Figure 3D: The figure legend is inappropriate.

9) Figure 3: In figure legend, “All data presented are mean ± SD of three similar independent experiments.” is unclear.

10) Figure 5B and D: The number of experiments is unclear.

11) Figure 5B: ** is unclear.

12) Figure 6B and D: * and ** are unclear.

13) Line 134-135: “A” series is not good compounds.

14) Some abbreviations are used without definition (e.g. “rt” in the figure legend of Scheme 1). Please avoid such inappropriate descriptions throughout the manuscript.

Author Response

Journal: International Journal of Molecular Sciences

Manuscript ID: ijms-2066139

Title: Discovery of tryptanthrin and its derivatives against NSCLC via autophagy promotes EGFR recycling and signaling

Author(s): Ya-Yu Zou 1,†, Guang-Long Zhang 2,†, Cheng-Peng Li 1, Hai-Tao Long 1, Dan-Ping Chen 1, Zhu-Rui Li 1, Gui-Ping Ouyang1,2,3, Wen-Jing Zhang 1, Yi Zhang 1, and Zhen-Chao Wang 1,2,3*

Dear Editor,

Thank you for giving us the opportunity to resubmit our manuscript. Based on your valuable comments and suggestions, we have revised it thoroughly. All suggestions and valued comments were considered in making changes. All queries raised by the reviewers have been addressed and remedial measures were adopted where necessary. Changes made in response to each comment are listed and discussed individually below. All changes refer to the previous version of the manuscript and all changes in our revised manuscript have been highlighted with red. The revised manuscript is hereby submitted to be considered for publication.

All changes refer to the previous version of the manuscript.

Open Review

English language and style

( ) English very difficult to understand/incomprehensible
(x) Extensive editing of English language and style required
( ) Moderate English changes required
( ) English language and style are fine/minor spell check required
( ) I don't feel qualified to judge about the English language and style

Reviewer: 1

Yes

Can be improved

Must be improved

Not applicable

Are all the cited references relevant to the research?

(x)

( )

( )

( )

Is the research design appropriate?

( )

(x)

( )

( )

Are the methods adequately described?

( )

(x)

( )

( )

Are the results clearly presented?

( )

(x)

( )

( )

Are the conclusions supported by the results?

( )

( )

(x)

( )

Comments and Suggestions for Authors

Zou and Zhang et al. developed derivative compounds that show anti-cancer activity based on tryptanthrin. Among them, the compound C1 showed the strongest activity, leading to apoptosis of cancer cell lines. The authors also demonstrated that the apoptosis would be caused by autophagy. Their results will be informative, but before consideration of publication in the International Journal of Molecular Science, the authors should respond to comments described below.

Specifics:

  • The title and contents are misleading. As shown in Tables 1 and 2, the compound C1 is not specific to A549 lung cancer cell line. This compound will affect to cell growth of a lot of types of cancer cell lines. Thus, the manuscript must be re-written without emphasizing lung cancer and EGFR signaling pathway.

Ans: Thanks for your kind suggestion. We quite agree with you that compound C1 shows a broad spectrum of anti-tumor activity and has no specificity for A549 cells. However, C1 has the best anti-tumor activity against A549 cells compared with other compounds. It is well known that EGFR is highly expressed in A549 lung cancer cell line, so we considering on lung cancer and EGFR signaling pathway in this article.

  • Related to 1), the results on EGF signaling pathway should be just one aspect among effects of C1. Therefore, it should be discussed in the Discussion section. In this context, Figure 6C and D should be shown in Supplementary Information. If the authors insist lung cancer and EGFR signaling pathway, they should add more convincing results on specific effects of C1 to lung cancer and EGFR signaling pathway.

Ans: Thanks for your kind suggestion. We quite agree that the result of EGFR signaling pathway is only one aspect of C1 effect. However, considering the higher expression of EGFR target in A549 cell line, we insist on lung cancer and EGFR signaling pathway. Indeed, in our study, the investigation of EGFR signaling pathway was convinced to our primal proposal. Meanwhile, we have explained the mechanism in the discussion part of the revised manuscript as follows.

Discussion

This study was conducted to assess the antitumor activity of tryptanthrin derivatives in NSCLC A549 cancer models in vitro, and to explore the important role of the autophagy in the antitumor effects of tryptanthrin. Autophagy is a major intracellular degradation system, and its degradation ability comes from lysosomes [7]. The new role of autophagy in regulating early endosomal homeostasis has been noticed by people. In the absence of autophagy, damaged or non-functional early endosomes accumulate EGFR and inhibit its recycling back to the plasma membrane. The results in the perturbation of (Epidermal growth factor) EGF-mediated signaling and survival and underlies a novel mechanism by which autophagy can regulate receptor tyrosine kinases (RTK) signaling [10]. EGFR is a well-studied receptor tyrosine kinase, which plays a vital role in regulating organ development and cancer progression [31]. Jane [10] et al. suggests that autophagy inhibition can reduce EGF-mediated signaling by prolonging the residence of EGFR on PI(3)P+ early endosomes. Therefore, autophagy can regulate the activity of some cellular cargoes such as receptor tyrosine kinases (RTKs), including the epidermal growth factor receptor (EGFR) and it has been proven to be a therapeutic target-for anticancer drug discovery, in particular for NSCLC treatment [32]. Interestingly, Peiyu et al. [33] found in their research that the occurrence of autophagy can be promoted by increasing the expression level of p-EGFR.

In the current study, we firstly demonstrated that anti-proliferation effects and clarify the possible molecular mechanisms of C1 on NSCLC. To evaluate the mechanism of cytotoxicity, apoptosis and cell cycle during treatment, further experiments on A549 cells were carried out by MTT assay and flow cytometry. Especially, the results showed that C1 induced apoptosis in A549 cells in a dose-dependent manner via up-regulating the level of Bax and reducing the expression of anti-apoptotic protein Bcl-2 and mitochondrial membrane potential reveals that C1 substantially decreased the level of mitochondrial membrane potential at the concentration of 2 μM. Meanwhile, it could arrest A549 cell cycle in the G2/M phase by down-regulating the expression of Cyclin D1 and this effect was also associated with the up-regulation of p21. Besides, cell migration showed that C1 affect cell motility in A549 cells. Autophagy plays an important role in maintaining early endosome function. Then we explored the effect of autophagy on the expression of EGFR tyrosine kinase key proteins. What's even more interesting, western blotting results revealed that the critical autophagy formation proteins LC3-II was significantly over expressed when exposed to the C1 at the concentration of 1 μM. While, the protein expression of the tumor suppressor gene p-EGFR and p-Akt was markedly up-regulated and the protein expression of EGFR and Akt was almost unchanged in A549 cells with treatment of C1. Based on the above results, we also found that the increase of p-EGFR protein expression can promote autophagy of cells, and excessive autophagy of cells can cause apoptosis.

3) C1 showed no effects at 1 μM but remarkable ones at 2 μM in some cases (e.g., Figures 4A, 4E, and 5A). In this regard, the cellular population at 2 μM are different from those at 0.25 - 1 μM (Figure 4A). Such drastic effects at 2 μM would reflect side-effects of C1. Additional experiments or convincing explanation would be necessary to avoid potential readers’ confusion.

Ans: Thanks for your kind suggestion. We really ensured the three parallel results of the data according to the scientific and rigorous nature of the experiment. Whether the concentration of C1 is 0.25-1 μM or 2 μM, we ensure that the collected cells are 10×104 living cells to analyze the data. In addtion, the influence of on cells is indeed much smaller at C1 concentration of 1 μM relative to 2 μM, which may not reflect side-effects of C1.

4)How about effects of C1 to normal cells?

Ans: The manuscript has been added the effect of compound C1 on human normal embryonic kidney cells. Unfortunately, the results showed that compound C1 has certain toxicity to human normal cells with the IC50 of 2.1 μM. Please see the supplementary materials for Table S1.

5) The English writing style is premature. I strongly recommend having a commercially available English editing service.

        Ans: Thanks for your kind suggestion very much. We have checked the grammar in the revised manuscript.

6) Scheme 1 should be included in a series of Figures.

Ans: Thanks for your kind suggestion. Scheme 1 has been included in a series of Figures in the current version of the manuscript. Therefore, the serial numbers of the following legends shall be extended by one digit.

  • Figure 3C: Concentration of C1 is unclear.

Ans: Thanks for your kind suggestion. The concentration of compound C1 in Figure 5C has been indicated in the manuscript, and it has also been indicated in the annotation.

  • Figure 3D: The figure legend is inappropriate.

Ans: Thanks for your kind suggestion. The figure legend has been properly modified in the Figure 5D in the manuscript.

  • Figure 3: In figure legend, “All data presented are mean ± SD of three similar independent experiments.” is unclear.

Ans: Thanks for your kind suggestion very much. We were sorry about this. The sentence of “All data presented are mean ± SD of three similar independent experiments.” is intended to explain Figure 5B in the revised version, which has been adjusted accordingly.

  • Figure 5B and D: The number of experiments is unclear.

Ans: Thanks for your kind suggestion. The number of experiments has been indicated in Figure 7B and D in the revised version. The sentence of “Data are expressed as the mean values ± standard deviations (SDs) of three different experiments (n = 3)” was added into the revised version.

  • Figure 5B: ** is unclear.

Ans: Thanks for your kind suggestion. “**” in Figure 7B of the revised version has been shown in the note, which indicates that there is a statistically extremely significant difference. At the same time, the “*” in Figure 6D of the revised version is also explained as “* Statistically significant (n = 3, p < 0.05).”

  • Figure 6B and D: * and ** are unclear.

Ans: Thanks for your kind suggestion. “* and **” in Figure 6B and D of the original version was modified with “* Statistically significant (n = 3, p < 0.05). ** Statistically extremely significant (n = 3, p < 0.01).” in Figure 8B and D of the current version of the manuscript.

  • Line 134-135: “A” series is not good compounds.

Ans: Thanks for your kind suggestion. I quite agree with you, “A” series is not good compounds. Line 134-135 has been modified in the manuscript.

  • Some abbreviations are used without definition (e.g. “rt” in the figure legend of Scheme 1). Please avoid such inappropriate descriptions throughout the manuscript.

Ans: Thanks for your kind suggestion. Some inappropriate abbreviations in the originnal manuscript has been checked and corrected in the revised version.

Reviewer 2 Report

The manuscript entitled “Discovery of tryptanthrin and its derivatives against non-small lung cancer via autophagy promotes EGFR recycling and signaling” is a worthy manuscript. However, there is a need for its improvement. Accordingly, the comments are provided below.

1. Title: “non-small lung cancer” may be written as “non-small cell lung cancer or NSCLC.”

2. Line 30: “non-small lung cancer” may be written as “non-small cell lung cancer or NSCLC.”

3. The authors must proofread the manuscript with Grammarly or Ginger software.  

4. The abbreviations must be written for the first time and in full form. After that, there is no need to write the full form (for example, lines 35-36 and 41 for NSCLC).

5. The words like “in vitro” must be italicized.

6. The authors can provide a pictorial mechanism of action of compound C1.

7. A pictorial representation of the SAR is advised.

8. The redundant part must be removed from the discussion part (for example, lines 257-260). The discussion parts should be specific to the results. Overall, a result-specific and literature-supported discussion must be written.

9. A theoretical ADME aspect of the C1 compound using free Swiss ADME software may also be discussed in the discussion part.

10. The conclusion part must also be rewritten specifically.

Author Response

Journal: International Journal of Molecular Sciences

Manuscript ID: ijms-2066139

Title: Discovery of tryptanthrin and its derivatives against NSCLC via autophagy promotes EGFR recycling and signaling

Author(s): Ya-Yu Zou 1,, Guang-Long Zhang 2,, Cheng-Peng Li 1, Hai-Tao Long 1, Dan-Ping Chen 1, Zhu-Rui Li 1, Gui-Ping Ouyang1,2,3, Wen-Jing Zhang 1, Yi Zhang 1, and Zhen-Chao Wang 1,2,3*

Dear Editor,

Thank you for giving us the opportunity to resubmit our manuscript. Based on your valuable comments and suggestions, we have revised it thoroughly. All suggestions and valued comments were considered in making changes. All queries raised by the reviewers have been addressed and remedial measures were adopted where necessary. Changes made in response to each comment are listed and discussed individually below. All changes refer to the previous version of the manuscript and all changes in our revised manuscript have been highlighted with red. The revised manuscript is hereby submitted to be considered for publication.

All changes refer to the previous version of the manuscript.

Open Review

English language and style

( ) English very difficult to understand/incomprehensible
( ) Extensive editing of English language and style required
( ) Moderate English changes required
( ) English language and style are fine/minor spell check required
(x) I don't feel qualified to judge about the English language and style

Yes

Can be improved

Must be improved

Not applicable

Does the introduction provide sufficient background and include all relevant references?

( )

(x)

( )

( )

Are all the cited references relevant to the research?

(x)

( )

( )

( )

Is the research design appropriate?

(x)

( )

( )

( )

Are the methods adequately described?

(x)

( )

( )

( )

Are the results clearly presented?

( )

(x)

( )

( )

Are the conclusions supported by the results?

( )

(x)

( )

( )

Comments and Suggestions for Authors

The manuscript entitled “Discovery of tryptanthrin and its derivatives against non-small lung cancer via autophagy promotes EGFR recycling and signaling” is a worthy manuscript. However, there is a need for its improvement. Accordingly, the comments are provided below.

Specifics:

  1. Title: “non-small lung cancer” may be written as “non-small cell lung cancer or NSCLC.”

Ans: Thanks for your kind suggestion. The suggestion has been adopted and the manuscript has been revised.

  1. Line 30: “non-small lung cancer” may be written as “non-small cell lung cancer or NSCLC.”

Ans: Thanks for your kind suggestion. The suggestion has been adopted and the manuscript has been revised. And, “non-small lung cancer” was written as “non-small cell lung cancer or NSCLC.”

  1. The authors must proofread the manuscript with Grammarly or Ginger software.

Ans: Thanks for your kind suggestion very much. We have checked the grammar in the revised manuscript.

  1. The abbreviations must be written for the first time and in full form. After that, there is no need to write the full form (for example, lines 35-36 and 41 for NSCLC).

Ans: Thanks for your kind suggestion. We have written the abbreviations for the first time and in full form.

  1. The words like “in vitro” must be italicized.

Ans: Thanks for your kind suggestion. The suggestion has been adopted and the manuscript has been revised.

  1. The authors can provide a pictorial mechanism of action of compound C1.

Ans: Thanks for your kind suggestion. The manuscript has been provided a pictorial mechanism of action of compound C1.

In conclusion, the comprehensive evaluation of the pharmacological effects of a number of novel tryptanthrin derivatives on tumor cells were performed in vitro. The an-tiproliferative activity of the products was screened against A549, K562, PC3, and HepG2 cell lines. Most derivatives performed moderate to significant potency, and the most promising C1 (IC50=0.55 ± 0.33 µM) exhibited excellent anti-proliferative activities against A549. Therefore, the representative compound C1 proves its anti-tumor mechanism. The structure–activity relationship (SAR) study indicated that replacing the A ring of tryptamine with pyridine ring may contribute the most to antitumor activity. Secondly, the anti-tumor activity of introducing electron-withdrawing substituent into B ring is better than that of introducing fatty amine group. C1 downregulates cell prolifer-ation and induces apoptosis and cell cycle arrest in A549 cells. Moreover, it was found that C1-induced apoptosis was caused by autophagy, which in turn affected the trans-mission of EGFR signal pathway. From these results, C1 could be considered as a promising anti-lung cancer chemotherapeutic for further evaluations.

Figure 9. The mechanism of compound C1.

  1. A pictorial representation of the SAR is advised.

Ans: Thanks for your kind suggestion. SAR relationship in the anti-tumor activity part of the manuscript has been described and a pictorial representation of the SAR has been provided.

Figure 4. The structure–activity relationship for tryptanthrin derivatives.

  1. The redundant part must be removed from the discussion part (for example, lines 257-260). The discussion parts should be specific to the results. Overall, a result-specific and literature-supported discussion must be written.

Ans: Thanks for your kind suggestion. The redundant part has been removed from the discussion part in the current manuscript, and the discussion parts have been rewritten.

  1. A theoretical ADME aspect of the C1 compound using free Swiss ADME software may also be discussed in the discussion part.

Ans: Thanks for your kind suggestion. A theoretical ADME aspect of the C1 were calculated using Swiss ADME software and ADMETlab 2.0 online websites and were discussed in the discussion part.

“More importantly, an successful candidate drug should been have a good ADME profile which includes absorption, distribution, metabolism, and excretion [34-36]. In this study, ADME profiles of the compound C1 were calculated using Swiss ADME software and ADMETlab 2.0 online websites. Unfortunately, the target compound C1 only has appropriate ADME properties (Figure S91 and Table S2). Our research group will further to optimize the structure of the lead compound and explore its action mechanism in the future.”

Figure S91. Predicted ADME properties of the target compound C1 .

Table S2. Pharmacokinetic (ADME) profiles of compound C1.

Compound

Absorption

Distribution

LogS

PCaco

%IA

P-gp substrate

Log VDss

%Fu

LogBB

BBB perm

Metabolism

Excretion tatal clearance log (ml/min/kg)

C1

-4.72

-4.741

<30%

Yes

0.668

9.4

/

Yes

CYP1A2 inhibitor

0.484

Note: The pharmacokinetic profiles are calculated in silico using SwissADME and ADMETlab 2.0 online websites. Abbreviations: % IA, predicted human intestinal absorption percent; BBB perm., BLOOD-brain barrier permeability; Fu: fraction unbound in plasma; logBB, predicted blood–brain partition coefficient; LogS, predicted aqueous solubility coefficient of a compound (log mol/L); PCaco, predicted apparent Caco-2 cell perme ability (log Papp in 10-6 cm/s).

  1. The conclusion part must also be rewritten specifically.

Ans: Thanks for your kind suggestion. The conclusion has been rewritten in the current manuscript.

Conclusions

In conclusion, the comprehensive evaluation of the pharmacological effects of a number of novel tryptanthrin derivatives on tumor cells were performed in vitro. The antiproliferative activity of the products was screened against A549, K562, PC3, and HepG2 cell lines. Most derivatives performed moderate to significant potency, and the most promising C1 (IC50=0.55 ± 0.33 µM) exhibited excellent anti-proliferative activities against A549. Therefore, the representative compound C1 proves its anti-tumor mechanism. The structure–activity relationship (SAR) study indicated that replacing the A ring of tryptamine with pyridine ring may contribute the most to antitumor activity. Secondly, the anti-tumor activity of introducing electron-withdrawing substituent into B ring is better than that of introducing fatty amine group. C1 downregulates cell proliferation and induces apoptosis and cell cycle arrest in A549 cells. Moreover, it was found that C1-induced apoptosis was caused by autophagy, which in turn affected the transmission of EGFR signal pathway. From these results, C1 could be considered as a promising anti-lung cancer chemotherapeutic for further evaluations.

Round 2

Reviewer 1 Report

The authors partially responded my questions. Their results and descriptions are still NOT convincing for publication in the International Journal of Molecular Science at this stage.

1) Again, the title and contents are misleading. The compound C1 is NOT specific to A549 lung cancer cell line. The compound C1 also affects to growth of the normal cell line (new Table S1). Thus, the manuscript MUST be re-written focusing on autophagy but NOT emphasizing lung cancer and EGFR signaling pathway to avoid readers' misunderstanding.

2) Related to 1), the results on EGFR signaling pathway should be just one aspect among effects of C1. If the authors insist lung cancer and EGFR signaling pathway, they MUST add more convincing results on specific effects of C1 to lung cancer and EGFR signaling pathway. What happen if phosphorylation of EGFR is blocked in the presence of C1? How about knockdown of EGFR? How about lung cancer cells showing lower levels of EGFR expression?

3) C1 showed no effects at 1 μM but remarkable ones at 2 μM in some cases (e.g., new Figures 6A, 6E, and 7A). Such drastic effects at 2 μM would reflect side-effects of C1. Additional experiments or convincing explanation is necessary in the main text to avoid potential readers’ confusion. In this regard, their responses were NOT convincing.

4) The English writing style is still premature. For example, line 230-231: “Data are expressed as the mean values ± standard deviations (SDs) of three different experiments (n = 3). ** Statistically extremely significant (n = 3, p < 0.01).” “SD” has been already used in Table 1. Why "SDs" but not "SD"? “n=3” was redundant. I recommend having a commercially available English editing service and showing a certificate.

Author Response

Journal: International Journal of Molecular Sciences

Manuscript ID: ijms-2066139

Title: Discovery of tryptanthrin and its derivatives against NSCLC via autophagy promotes EGFR recycling and signaling

Author(s): Ya-Yu Zou 1,†, Guang-Long Zhang 2,†, Cheng-Peng Li 1, Hai-Tao Long 1, Dan-Ping Chen 1, Zhu-Rui Li 1, Gui-Ping Ouyang1,2,3, Wen-Jing Zhang 1, Yi Zhang 1, and Zhen-Chao Wang 1,2,3*

Dear Editor,

Thank you for giving us the opportunity to resubmit our manuscript. Based on your valuable comments and suggestions, we have revised it thoroughly. All suggestions and valued comments were considered in making changes. All queries raised by the reviewers have been addressed and remedial measures were adopted where necessary. Changes made in response to each comment are listed and discussed individually below. All changes refer to the previous version of the manuscript and all changes in our revised manuscript have been highlighted with red. The revised manuscript is hereby submitted to be considered for publication.

All changes refer to the previous version of the manuscript.

Review Report Form

Open Review

English language and style

( ) English very difficult to understand/incomprehensible
(x) Extensive editing of English language and style required
( ) Moderate English changes required
( ) English language and style are fine/minor spell check required
( ) I don't feel qualified to judge about the English language and style

Yes

Can be improved

Must be improved

Not applicable

Does the introduction provide sufficient background and include all relevant references?

(x)

( )

( )

( )

Are all the cited references relevant to the research?

(x)

( )

( )

( )

Is the research design appropriate?

( )

(x)

( )

( )

Are the methods adequately described?

( )

(x)

( )

( )

Are the results clearly presented?

( )

(x)

( )

( )

Are the conclusions supported by the results?

( )

( )

(x)

( )

Comments and Suggestions for Authors

The authors partially responded my questions. Their results and descriptions are still NOT convincing for publication in the International Journal of Molecular Science at this stage.

  • Again, the title and contents are misleading. The compound C1 is NOT specific to A549 lung cancer cell line. The compound C1 also affects to growth of the normal cell line (new Table S1). Thus, the manuscript MUST be re-written focusing on autophagy but NOT emphasizing lung cancer and EGFR signaling pathway to avoid readers' misunderstanding.

Ans: Thanks for your kind suggestion. We quite agree with you that compound C1 shows a broad spectrum of anti-tumor activity and has no specificity for A549 cells. Therefore, according to your valuable comments and suggestions, we have thoroughly revised the manuscript. In the current version of the manuscript, the focus is on autophagy, without overemphasizing lung cancer and EGFR signaling pathways. In this context, the original Figure 8C and D have been shown in Supplementary Information (Figure S92). In addition, the title of this article has also been revised as “Discovery of tryptanthrin and its derivatives against NSCLC in vitro via both apoptosis and autophagy pathways”.

  • Related to 1), the results on EGFR signaling pathway should be just one aspect among effects of C1. If the authors insist lung cancer and EGFR signaling pathway, they MUST add more convincing results on specific effects of C1 to lung cancer and EGFR signaling pathway. What happen if phosphorylation of EGFR is blocked in the presence of C1? How about knockdown of EGFR? How about lung cancer cells showing lower levels of EGFR expression?

Ans: Thanks for your kind suggestion very much. In the end, we didn't insist on lung cancer and EGFR signaling pathway.

  • C1 showed no effects at 1 μM but remarkable ones at 2 μM in some cases (e.g., new Figures 6A, 6E, and 7A). Such drastic effects at 2 μM would reflect side-effects of C1. Additional experiments or convincing explanation is necessary in the main text to avoid potential readers’ confusion. In this regard, their responses were NOT convincing.

Ans: Thanks for your kind suggestion. In Figure 6A, compared with the control group, the total apoptosis rates (early and late apoptotic cells) of A549 cells treated with C1 in 1 μM were 92.89%. In the study of the influence of C1 on cell cycle distribution, after 24 hours of treatment with C1 (1 and 2 μM), the percentage of cells in G2/M increased to 24.1% and 74.5%. However, the results of Figure 6E clearly showed that the mitochondrial membrane potential drops sharply or even collapses at the concentration of C1 in 2 μM, whereas these remarkable features are almost absent in A549 lung cancer cells of control and other concentrations. At the concentration of 2 μM, such a drastic effect is suspected to be caused by excessive autophagy of cells, rather than the side effect of C1. Because autophagy also occurred at 2 μM, observed by electron microscope. Therefore, we conducted additional experiment to illustrate it. The normal cell line HEK-293 were treated with C1 and observed the cell proliferation and death by DAPI staining. The results showed that the toxic and side effects were general and acceptable at the concentration of C1 2 μM, and the explanation was also given in the text.

Figure S93. Effect of C1 on the normal cell line HEK-293. Scale bars are 100 μm.

  • The English writing style is still premature. For example, line 230-231: “Data are expressed as the mean values ± standard deviations (SDs) of three different experiments (n = 3). ** Statistically extremely significant (n = 3, p < 0.01).” “SD” has been already used in Table 1. Why "SDs" but not "SD"? “n=3” was redundant. I recommend having a commercially available English editing service and showing a certificate.

Ans: Thanks for your kind suggestion very much. We have checked the grammar in the revised manuscript.

Round 3

Reviewer 1 Report

The authors responded most part of my questions.

Comments:

1)      This study examined (focuses on) anti-tumor effects of specific chemical compounds (C1 etc.) but not mechanisms of autophagy. Autophagy is just a part of consequence. Thus, it would be confusing that autophagy is described in the first paragraph of the Introduction. The contents of the first paragraph of the Introduction should be included in the first paragraph in the Discussion section.

2)      The magnification of microscopy (photos) in Figure S93 should be the same as Figure 5D.

3)      Supplementary Figures S91-93 appears in the wrong order (S93, S92, S91) in the main text. Please correct the order (S91, S92, S93).

4)      No citation on Figure 9 in the main text.

5)    English writing is still premature so that it is stressful to read for me and potential readers… Please use a commercially available English editing service and show an issued certificate. In this regard, the contents of this manuscript are ordinary but never outstanding. However, the results might be informative for researchers working in the same filed. Therefore, I strongly recommend improving English writing so that readers can appropriately understand the contents. If English writing is terrible, neither the contents of this manuscript nor this journal will be trusted by readers.

Author Response

The authors responded most part of my questions.

Comments:

  • This study examined (focuses on) anti-tumor effects of specific chemical compounds (C1 etc.) but not mechanisms of autophagy. Autophagy is just a part of consequence. Thus, it would be confusing that autophagy is described in the first paragraph of the Introduction. The contents of the first paragraph of the Introduction should be included in the first paragraph in the Discussion section.

Ans: Thanks for your kind suggestion. Autophagy is really only a part of the consequences. Thus, the content in the first paragraph of the introduction has been included in the first paragraph of the discussion section.

2)      The magnification of microscopy (photos) in Figure S93 should be the same as Figure 5D.

Ans: Thanks for your kind suggestion. We were sorry about our mistake. The Figure S93 should be named as S91. The microscope (photo) in Figure S91 has been photographed again. Now, the magnification of the microscopy (photo) in Figure S91 has been made the same as that in Figure 5D.

Figure S91. Effect of C1 on the normal cell line HEK-293. The cell morphology was observed under the fluorescence microscope (20x). Scale bars are 100 μm.

3)      Supplementary Figures S91-93 appears in the wrong order (S93, S92, S91) in the main text. Please correct the order (S91, S92, S93).

Ans: Thanks for your kind suggestion. The order in which supplementary Figures S91-93 appears in the text has been corrected.

4)      No citation on Figure 9 in the main text.

Ans: Thanks for your kind suggestion. Figure 9 has been cited in the revised text.

5)    English writing is still premature so that it is stressful to read for me and potential readers… Please use a commercially available English editing service and show an issued certificate. In this regard, the contents of this manuscript are ordinary but never outstanding. However, the results might be informative for researchers working in the same filed. Therefore, I strongly recommend improving English writing so that readers can appropriately understand the contents. If English writing is terrible, neither the contents of this manuscript nor this journal will be trusted by readers.

Ans: Thanks for your kind suggestion. We have checked the grammar in the revised manuscript and marked it with red colour.
